# The Baron Pasquale Revoltella’s Will in the Forensic Genetics Era

**DOI:** 10.3390/genes14040851

**Published:** 2023-03-31

**Authors:** Paolo Fattorini, Carlo Previderè, Serena Bonin, Solange Sorçaburu Ciglieri, Pierangela Grignani, Paola Pitacco, Monica Concato, Barbara Bertoglio, Irena Zupanič Pajnič

**Affiliations:** 1Department of Medicine, Surgery and Health, University of Trieste, 34149 Trieste, Italy; 2Department of Public Health, Experimental and Forensic Medicine, Section of Legal Medicine and Forensic Sciences, University of Pavia, 27100 Pavia, Italy; 3Institute of Forensic Medicine, Faculty of Medicine, University of Ljubljana, 1000 Ljubljana, Slovenia

**Keywords:** skeletal remains, PCR-MPS, SNP, mtDNA, DNA degradation

## Abstract

In this article, we describe multiple analytical strategies that were first developed for forensic purposes, on a set of three bone samples collected in 2011. We analyzed a single bone sample (patella) collected from the artificially mummified body of the Baron Pasquale Revoltella (1795–1869), as well two femurs which allegedly belonged to the Baron’s mother (Domenica Privato Revoltella, 1775–1830). Likely due to the artificial mummification procedures, the inner part of the Baron’s patella allowed the extraction of high-quality DNA yields, which were successfully used for PCR-CE and PCR-MPS typing of autosomal, Y-specific, and mitochondrial markers. The samples extracted from the trabecular inner part of the two femurs yielded no typing results by using the SNP identity panel, whereas the samples extracted from the compact cortical part of the same bone samples allowed genetic typing, even by the employment of PCR-CE technology. Altogether, 10/15 STR markers, 80/90 identity SNP markers, and HVR1, HVR2, and HVR3 regions of the mtDNA were successfully typed from the Baron’s mother’s remains by the combined use of PCR-CE and PCR-MPS technologies. The kinship analysis showed a likelihood ratio of at least 9.1 × 10^6^ (corresponding to a probability of maternity of 99.9999999%), and thus confirmed the identity of the skeletal remains as those of the Baron’s mother. This casework represented a challenging trial for testing forensic protocols on aged bones samples. It highlighted the importance of accurately sampling from the long bones, and that DNA degradation is not blocked by freezing at −80 °C.

## 1. Introduction

The recent developments in molecular biology allow analyses that were inconceivable until only a few decades ago. Among these extraordinary advances, scientists from different branches are now able to perform studies on ancient or museum specimens [1,2].

In forensics, DNA profiling is a routine tool in criminal investigations [3], and skeletal remains are one of the most challenging samples [4]. Autosomal STR (short tandem repeat) typing is the gold standard for individual identification in forensics [3,5], however, in some circumstances other DNA markers can be used. For example, iSNPs (identity single nucleotide polymorphisms) and InDel (insertion/deletion) typing may be greatly beneficial when analyzing degraded samples [3,6,7]. It is also possible, however, that DNA degradation [8], an unavoidable process of the post mortem tissues, can make nuclear genetic testing of scarce practical utility, or even inconclusive [3,5,8]. In such cases, the analysis of mitochondrial DNA (mtDNA) can be of help [9].

Capillary electrophoresis (CE) analysis of PCR (polymerase chain reaction) products is the gold standard approach in forensics [3,5], whereas MPS (massive parallel sequencing) is an emerging promising technology in the typing of low-template degraded samples [10,11,12]. Although high-throughput shotgun sequencing and the analysis of genome-wide data have largely replaced current PCR-based methods in ancient DNA (aDNA) analysis [1,2,13], there are many aspects that aDNA analysis and forensic DNA analysis have in common; for instance, the use of limited amounts of degraded DNA, the precautions adopted to prevent contamination, and the use of authenticity criteria [14,15,16,17,18,19,20]. In addition, both disciplines have developed strategies to select the skeletal element which provides, a priori, the highest probabilities of positive outcomes [21,22,23,24,25,26,27,28,29,30,31,32,33,34,35,36,37].

In the last decade, several genetic studies have been conducted on the skeletal remains of famous figures from the past, such as Nicolaus Copernicus [38] and King Richard III [39], as well from lesser-known individuals [40,41] or mass graves victims of the Spanish Civil War [42] and Second World War [43]. These studies, conducted by interdisciplinary teams of geneticists, archaeologists, anthropologists, and historians, were excellent trials to assess the performance of standard and emerging technologies, as well as solving historical and archeological questions [38,39,40,41,42,43,44,45,46].

In the early 19th century, Baron Pasquale Revoltella lived in Trieste, Italy (see Appendix B for his short biography and other historical details), and he left the following will: “I do not like being buried as I am, I want to be embalmed in the Egyptian manner … and then to be laid in the ready-made sarcophagus in the Crypt of the Church of San Pasquale erected in the park of my country villa. My tomb will be reopened one hundred years after my death, and will be closed after three days of unforgettable celebrations”. Following his will, his body was mummified and laid in the crypt, but neither exhumation nor celebrations were carried out in 1969 (i.e., one hundred years after his death). However, one hundred and forty-two years after his death, his body was exhumed and a single bone element was sampled. In addition, bone remains that likely belong to the Baron’s mother were sampled to confirm the identity. The results of the molecular analyses that follow the forensic protocols are shown and discussed below.

## 2. Materials and Methods

### 2.1. Sample Collection and Precautions to Avoid Contaminations

The body of the Baron Revoltella was exhumed from the crypt of the San Pasquale Church on 4 June 2011. The body lay inside a metal coffin. The examination of the body was carried out the same day, and the body showed clear signs of complete artificial mummification. The collection of a single bone sample was allowed. Therefore, after cutting the treasures and the bandages, the whole right patella (sample P) was excised (see Figure A1). This bone sample was inserted in a sterile tube and transferred to the laboratory where it was processed immediately. After sawing (see Figure A2), a few specimens were selected for histological analysis, whereas the remaining portion was stored in a sealed tube at −80 °C until the molecular analyses.

In the course of the exhumation procedure, a metal box with the inscription “Domenica Revoltella”, Baron’s mother’s name, was found in a niche next to the Baron grave. The metal box contained bone remains whose anthropological examination revealed the presence of an incomplete unique female skeleton; the skeleton was of a female individual, 55–65 years old, approximately 159–161 cm of height. Therefore, it was hypothesized that these skeletal remains belonged to the Baron’s mother, Domenica Privato Revoltella (D.P.R.). For molecular purposes, as no tooth was found, one segment of about ten centimeters each was sawn from the diaphysis of the right femur (sample RF) and the left femur (sample LF) (see Figure A3). These samples were then transferred in sterile tubes and stored in the dark at room temperature until the time of the molecular analyses.

Throughout the procedures, precautions were taken to avoid modern DNA contamination [16,17,18,19,47]. For the elimination database, buccal swabs were obtained after informed consent from all personnel involved in these operations, as well in the molecular analyses. All methods, including genetic data storage, were performed in accordance with the guidelines and regulations of the Ethics Committee of the University of Trieste (101/04.12.2019).

### 2.2. DNA Extraction and Quantification

The extraction procedures were carried out in rooms dedicated solely to aged bones analysis, and adopting stringent precautions to prevent contamination [47]. DNA from samples P1, LF1, and RF1 were isolated in 2012 as previously described [48] with minimal modifications. Briefly, 0.5 g of the inner (trabecular) part of each bone was crushed in a mortar, and decalcified in 5 mL of 0.5 M Na_2_EDTA at room temperature for 48 h. After centrifugation, the pellet was resuspended in 5 mL of lysis buffer with proteinase K (at a final concentration of 200 µg/mL) and incubated at room temperature for 24 h. After two phenol/chloroform/isoamyl alcohol (25/24/1) purifications, and one chloroform/isoamyl alcohol purification, the extract was filtered through a K100 Amicon column. After three washes with water and one with low TE buffer (LTE; 1 mM Tris pH 7.6, 0.1 mM Na_2_EDTA pH 8.0), 30–35 µL of extract was obtained. Negative extraction controls (NEC) were carried out simultaneously. The extract was aliquoted and stored at −20 °C until use. DNA quantification was carried out by the use of the Quantifiler^TM^ Trio DNA Quantification kit (Thermo Fisher Scientific, Waltham, MA, USA).

As shown in Table 1, samples P2, RF2, and LF2 were extracted in 2022 as previously reported [47] (for femurs, compact cortical parts were used). Briefly, 0.5 g of the powdered bone was incubated in 0.5 M Na_2_EDTA at 37 °C overnight. After centrifugation, the pellet was washed with water and extracted with the EZ1 DNA Investigator Kit (Qiagen, Hilden, Germany) in a final volume elution of 50 µL. A Biorobot EZ1 device (Qiagen) was used. NECs were carried out simultaneously. DNA quantification of samples RF2 and LF2 was performed using the PowerQuant Kit (Promega, Madison, WI, USA).

The number of mtDNA molecules was assessed in samples P1, P2, and LF2 using an in-house qPCR method based on the study of Alonso et al. [49]. Accordingly, a 620 bp long fragment as standard was used, and 113 bp and 287 bp fragments of mitogenome defined by two custom-ordered primer solutions (Applied Biosystems), Renfrewshire, UK) were quantified, using the QuantStudio™ 5 Real-Time PCR System (Thermo Fisher Scientific, Waltham, MA, USA) and Design and Analysis Software v1.5.2 (Thermo Fisher Scientific, Waltham, MA, USA). A longer, 287 bp fragment was also quantified to estimate the mitogenome degradation. The mtDNA degradation was calculated for a given sample by the ratio between the 113 bp mtDNA copies and the 287 bp mtDNA copies. All samples were quantified in duplicate. In lieu of 1X TaqMan Universal PCR Master Mix (Applied Biosystems, Renfrewshire, UK), which was used in the initial study [49] but was not available anymore, a newer commercially available alternative was used: the TaqMan™ Universal Master Mix II with Uracil-N-glycosylase (Thermo Fisher Scientific, Waltham, MA, USA).

The molecular weight of sample P2 was assessed by 1.2% agarose gel electrophoresis in TBE buffer containing EtBr (5 ng/mL).

### 2.3. DNA Typing

As shown in Table 2, different approaches were carried out for genetic typing. They can be summarized as follows.

#### 2.3.1. STR-Y Typing by PCR-CE

Approximately 0.8 ng of DNA from sample P1 was amplified, in duplicate tests, with 30 PCR cycles using the Y-filer Kit (Applied Biosystems, Renfrewshire, UK). Amplicons were typed by CE analysis in a 310 ABI Prism apparatus (Applied Biosystems, Renfrewshire, UK). NECs and no template controls (NTCs) were tested simultaneously.

#### 2.3.2. Autosomal STR Typing by PCR-CE

For samples RF2 and LF2, 17.5 µL of DNA solution corresponding to 129 and 194 pg, respectively, were amplified with 30 PCR cycles using the PowerPlex ESX Kit (Promega, Madison, WI, USA). Five-hundred picograms of sample P1 DNA were tested in duplicate (30 PCR cycles). Amplicons were typed by CE analysis in a 310 ABI Prism or a SeqStudio apparatus (Thermo Fisher Scientific, Waltham, MA, USA). NEC and NTE were analyzed simultaneously.

#### 2.3.3. iSNP Typing by PCR-MPS

The HID-identity panel (allowing the analysis of 90 autosomal iSNPs plus 34 Y-specific SNPs) was used in two different experiments with different techniques. Briefly, libraries from samples LF1, RF1, and P1 were built manually [50]. One nanogram of P1 DNA was amplified in duplicate with 21 PCR cycles, whereas 15 µL of DNA from samples LF1 and RF1 were amplified with 25 PCR cycles. In the second experiment, 15 µL of DNA from samples RF2 and LF2, corresponding to 111 and 166 pg, respectively, were amplified with 27 PCR cycles using the Ion Chef apparatus (Thermo Fisher Scientific, Waltham, MA, USA). Sample RF2 was run in duplicate. Libraries at the concentration of 30 pM were pooled and run in a chip using an Ion apparatus (Thermo Fisher Scientific, Waltham, MA, USA) [51]. NEC and NTE were analyzed simultaneously. The analytical threshold of 50 reads was applied for locus call [50].

#### 2.3.4. mtDNA Analysis

Both PCR-MPS and PCR-CE approaches were performed (see Table 2) for mtDNA typing.

For PCR-CE analysis, mtDNA hypervariable regions I and II (HVR-I and HVR-II) were amplified separately, as previously described [52], in a final volume of 25 µL, with 35 PCR cycles. To each sample, 1 U GoTaq^®^ Flexi DNA Polymerase (Promega, Madison, WI, USA) and 500 pg DNA recovered from samples RF2 and P1 were added. The molecular weight of the amplified products was checked by agarose gel electrophoresis. PCR products were then purified using QIAquick PCR NucleoSpin Gel (Qiagen, Hilden, Germany) following the manufacturer’s instructions. Sanger sequencing reactions were carried out using the BigDye Terminator v3.1 Cycle Sequencing Kit (Applied Biosystems, Renfrewshire, UK) with primers (forward and reverse) used for the amplification reactions. Unincorporated dye terminators were removed from the reaction using the NucleoSEQ Kit (Macherey-Nagel, Dueren, Germany). Sequences were separated by capillary electrophoresis on a SeqStudio Genetic Analyzer (Thermo Fisher Scientific, Waltham, MA, USA). Raw data were analyzed using the Sequencing Analysis v.5.2 software, and the resulting electropherograms were compared with the rCRS (revised Cambridge reference sequence) [53]. The mtDNA haplotypes were then checked for quality parameters in the EMPOP database [54,55] and for the phylogenetic assignment of the corresponding haplogroup.

For PCR-MPS analysis, automated combined library preparation was carried out on an HID Ion Chef™ Instrument (Thermo Fisher Scientific, Waltham, MA, USA) with the Precision ID DL8 Kit™ (Thermo Fisher Scientific, Waltham, MA, USA) and Precision ID mtDNA Control Region Panel (Thermo Fisher Scientific, Waltham, MA, USA), following the manufacturer’s instructions [56]. Accordingly, for samples LF2, P1, and P2 approximately 10,000 mtDNA molecules were used for each sample. The number of primer pools was 2, the number of PCR cycles was 22, and anneal and extension times were 4. Each combined library was quantified in duplicate with the Ion Library TaqMan™ Quantitation Kit (Thermo Fisher Scientific, Waltham, MA, USA) in a QuantStudio™ 5 Real-Time PCR System (Thermo Fisher Scientific, Waltham, MA, USA) following the manufacturer’s guidelines [56]. Raw data were analyzed with Design and Analysis Software v1.5.2 (Thermo Fisher Scientific, Waltham, MA, USA). Equimolar amounts required for superpooling libraries were calculated as recommended by the manufacturer. Templating was fully automated by the use of an Ion 530™ Chip (Thermo Fisher Scientific, Waltham, MA, USA) in an Ion Chef™ Instrument (Thermo Fisher Scientific, Waltham, MA, USA), with dedicated reagents, namely, the Ion S5™ Precision ID Chef Supplies, the Ion S5™ Precision ID Chef Reagents, and the Ion S5™ Precision ID Chef Solutions (Thermo Fisher Scientific, Waltham, MA, USA), following the manufacturer’s recommendations (Thermo Fisher Scientific, Waltham, MA, USA, 2021). Accordingly, 30 pM of each pool was used for templating. The Ion GeneStudio™ S5 System (Thermo Fisher Scientific, Waltham, MA, USA), together with Ion S5™ Precision ID Sequencing Reagents and Ion S5™ Precision ID Sequencing Solutions (Thermo Fisher Scientific, Waltham, MA, USA), were used to generate raw data for mtDNA sequencing. Primary analysis of raw data, including sequence alignment to rCRS and variant calling, was performed with the Ion Torrent™ Suite 5.10.1 (Thermo Fisher Scientific, Waltham, MA, USA) software and HID Genotyper 2.2 and Coverage Analysis (v5.10.0.3) plugins. Secondary data analysis was carried out with Converge™ Software v2.3 (Thermo Fisher Scientific, Waltham, MA, USA).

### 2.4. Histological Examination

To analyze the preservation of bone tissue, bone thin sections were cut from the patella bone, fixed in 10% neutral buffered formalin (ratio formalin/sample 20:1), and decalcified at room temperature with 0.5 M Na_2_EDTA pH 8.0. After dehydration in increasing alcohol series and xylene, samples were embedded in paraffin, and 5 μm sections were cut and stained with Hematoxylin–Eosin following standard procedures.

Bone tissue was analyzed and classified according to the Oxford histological index (OHI), as previously described [57,58]. Six stages were defined (from 0 to 5), considering the amount of well-preserved bone tissue and the possible identification of bone features, such as osteons, lamellae, and osteocyte lacunae. Well-preserved bone tissue, comparable to fresh bone, with more than 95% of intact bone, was classified as “5”, while bone sections with no recognizable features and less than 5% of well-preserved bone tissue were classified as “0”.

### 2.5. Data Analysis

Consensus methods for results interpretation were adopted, when duplicate tests were carried out. For STR data, methods described by Taberlet et al. [59] were used, whereas for SNP data, the method described by Turchi et al. [50] was used. The YHRD database (https://yhrd.org/) was used to analyze the Y-STR profile obtained from the Baron’s sample. For kinship analysis, the Familias software (version 3.2.9) was used. As reference databases, the STR allele frequencies of Italian population [60] and the SNP allele frequencies of Caucasian population (https://www.ncbi.nlm.nih.gov/snp) were used. For Y haplogroup prediction of sample P1, the plugin HID SNP Genotyper, as well the websites http://ytree.morleydna.com and http://phylotree.org/Y/tree/index.htm, were used. To check mtDNA haplotype frequencies and the corresponding haplogroup prediction, the website EMPOP database [54] was used.

## 3. Results

### 3.1. DNA Isolation and Quantification

Quantification of autosomal DNA from the Baron’s sample (sample P1) returned high DNA quantities with a degradation index of 1.8, indicative of minimal degradation (see Table 2). The good preservation of the sample was also supported by agarose gel electrophoresis, as shown in Figure 1.

The results of the quantification of the Baron’s mother samples are shown in Table 1. Three out of four samples revealed detectable levels by qPCR for the short autosomal targets.

**Table 1 genes-14-00851-t001:** List of the samples in this study. D.P.R.: Domenica Privato Revoltella; Year: year of the DNA extraction; S.A.T.: concentration of the sample as assessed by the short autosomal target in the qPCR assay; L.A.T.: concentration of the sample as assessed by long autosomal target in the qPCR assay; D.I: ratio between S.A.T. and L.A.T. The last two columns show the number of mitochondrial (mt) DNA molecules, as assessed by the 113 bp and 287 bp long targets, respectively. LOD: limit of detection; n-c: not calculable; n.p.: not performed.

Donor	Specimen	Sample #	Year	S.A.T. (ng/µL)	L.A.T. (ng/µL)	D.I.	mt (113 bp)	mt (287 bp)
Baron	Patella (trabecular)	P1	2012	0.2861	0.1582	1.8	349,685	20,105
		P2	2022	n.p.	n.p.	/	2060	<LOD
D.P.R.	Right femur (trabecular)	RF1	2012	<LOD	<LOD	n-c	n.p.	n.p.
	Right femur (compact)	RF2	2022	0.0074	0.0006	11.6	n.p.	n.p.
	Left femur (trabecular)	LF1	2012	0.0042	<LOD	n-c	n.p.	n.p.
	Left femur (compact)	LF2	2022	0.0111	0.0011	10.3	20,441	<LOD

DNA isolated from the outer part (compact cortical bone tissue) of the two femurs had a degradation index (D.I.) of 11.6 and 10.3, respectively, supporting better preservation of the outer part samples compared to the inner part of the femurs (trabecular bone tissue). No amplification was obtained for the long targets in DNA samples from the trabecular part of the two bones (samples RF1 and LF1).

Mitochondrial DNA copies in samples LF2, P1, and P2 are reported in Table 1. It is noteworthy that the number of mtDNA copies/g of tissue in sample P2, which was stored at −80 °C for ten years, was about two orders of magnitude lower than in sample P1 (extracted in 2012). Furthermore, while from sample P1 it was possible to amplify both targets (with a ratio of approximately 17 between the short and the long target), in sample P2 no amplification of the longer target (287 bp) was obtained. Altogether, those data strongly point out that DNA degradation occurred during the bone sample storage at −80 °C. No mtDNA was detected in the NEC.

### 3.2. DNA Typing

The overall DNA typing results were summarized in Table 2.

**Table 2 genes-14-00851-t002:** Results of the DNA typing by CE (PCR-CE-based systems) and MPS (PCR-MPS-based systems); +: positive outcome (more than 50% of the markers); -: no result; n.p.: not performed.

Sample	Y-Specific STRs		Autosomal STRs		Mitochondrial Control Region	
	CE	MPS	CE	MPS	CE	MPS
P1	+	+	+	+	+	+
P2	n.p.	n.p.	n.p.	n.p.	n.p.	+
RF1	/	/	n.p.	-	n.p.	n.p.
RF2	/	/	+	+	+	n.p.
LF1	/	/	n.p.	-	n.p.	n.p.
LF2	/	/	+	+	n.p.	+

The detailed results are hereafter presented.

#### 3.2.1. Y-STR Typing by PCR-CE

Results of Y-STR typing in sample P1 are reported in the Appendix A. The duplicate test results confirmed the genetic data obtained from DNA isolated a few weeks after the exhumation. No amplification was obtained from negative controls. The obtained haplotype did not match with the entire HYRD database (searching among 290,147 haplotypes; accessed on 26 January 2023). Furthermore, no match was found with haplotypes from the male staff members involved in the study. Overall, these results, supported by the high amount of amplifiable DNA obtained from the patella bone, were used as authenticity criteria [15,25]. The “minimal haplotype” analysis, performed on eight markers of 350,500 unrelated samples, showed 202 matches (mainly in the USA with 22 matches, followed by Spain and Italy).

#### 3.2.2. Autosomal STR Typing by PCR-CE

The PowePlex ESX Kit was used with 30 PCR cycles. Results of the analysis for samples RF2 and LF2 are reported in Appendix A (see also Figure A4). No amplicon was detected in the negative controls. Genetic typing data were used to generate a *consensus* profile for 10 out of 16 loci [59]. A full profile was achieved from the Baron’s sample in duplicate tests. By comparing the Baron’s profile with that of the alleged mother D.P.R., allele sharing was always scored, as shown in Appendix A. Familias software returned a likelihood ratio (LR) value of maternity of 3409 (corresponding to a posterior probability of maternity of 99.971%).

#### 3.2.3. iSNP Typing by PCR-MPS

Samples RF1, LF1, and P1 results were previously reported [50]. Briefly, a full and complete profile was achieved from the Baron’s sample, while samples RF1 and LF1 (both extracted from the trabecular bone tissue) did not return any result. On the opposite, libraries of good quality were obtained from RF2 and LF2 DNA samples extracted from the compact cortical bone tissue of the two femurs (Appendix A for sequencing parameters). Possibly due to the higher number of PCR cycles (27), a high coverage of the markers was obtained, with more than 300,000 mapped reads/library. Genotyping results are reported in Appendix A. No result was obtained from NEC. The three independent tests data allowed generating the *consensus* profiles [50] for 80 out of 90 autosomal identity SNPs. Ten markers did not return reliable results, as shown in Appendix A, because of stochastic phenomena (locus drop out and allelic drop out) which were scored in the replicates. However, when the *consensus* profile was compared with the sample P1 genotype, allele sharing was always scored. Furthermore, Familias software returned a likelihood ratio (LR) value of maternity of 2678 (corresponding to a posterior probability of 99.962%). When SNP and STR typing results were computed together, the cumulative LR value was of 9,129,302 (corresponding to a probability of maternity of 99.9999).

Data from the 34 Y-specific SNP markers of the identity panel (see Appendix A) were used to establish the Baron’s ancestry. The “Y Haplogroup Prediction” made with the plugin of the HID SNP Genotyper revealed that the Baron’s sample belongs to the R1b (R1b-M343) haplogroup. This prediction was further confirmed by the use of http://ytree.morleydna.com and Phylo Tree Y (http://phylotree.org/Y/tree/index.htm) sites (accessed on 26 January 2023).

#### 3.2.4. mtDNA Typing

Mitochondrial DNA typing in samples P1, P2, RF2, and LF2 was carried out with two technologies, namely, PCR-CE and PCR-MPS (see Table 1).

Hypervariable control regions HVR-I and HVR-II were amplified and sequenced by conventional Sanger sequencing, while for MPS, the entire hypervariable control region was investigated by the addition of the HVR-III hypervariable region (see Appendix A for sequencing parameters). The resulting haplotypes are reported in Appendix A.

Both methods returned the same haplotype for HVR-I and II in all tested samples. Poly-C stretch length variation at np 309 was observed in all tested samples with both techniques (dominant variant 309.1 C), with a visible additional C insertion (309.2 C) identified only in samples P1 and RF2 by Sanger sequencing. This additional C insertion was not recorded in the three samples analyzed by MPS. Furthermore, a point heteroplasmy at np 16,168 (16168 Y) was detected only in sample LF2 by the MPS technology. Those variations can likely be due to the different DNA sources (RF and LF for D.P.R. samples) or to the different technologies and chemistries used [55]. Despite this, the genetic compatibility of the haplotypes supports the maternal relationship between the Baron and D.P.R.

To check the frequency of the haplotypes in the population and for haplogroup estimation, the haplotypes were loaded in the EMPOP database [54]. InDels at np 309 were disregarded, and ranges corresponding to the sequenced mitochondrial region were established (Sanger sequencing: 16024-16365, 60-340; MPS technology: 16024-576). Matches were observed in the worldwide and European databases, providing haplotype frequencies between 1 × 10^−2^ and 1 × 10^−4^. Phylogenetic analyses assigned the samples to haplogroup HV0 (Sanger sequencing data) and V18 (MPS data), both frequent in the European population. In particular, haplogroup V18 is frequent in the Netherlands, Germany, and Italy (https://www.eupedia.com/europe/Haplogroup_V_mtDNA.shtml, accessed on 13 February 2023). The assignment of those different haplogroups was related to the identification of the additional mutation in the HVR-III region (508G) by the MPS method. Since the sequence information from the entire control region is associated to a greater resolution of the phylogenetic tree [55], the haplotype obtained from MPS analysis, and therefore haplogroup V18, was considered as a final result.

### 3.3. Histological Examination

The visual examination of the Baron’s patella showed a good preservation of the bone tissue. This finding was confirmed by microscopic histological investigation as well. All sections showed intact bone (more than 95% of the tissue) with well-recognizable bone tissue components, such as bone canals, lamellae, and osteocyte lacunae. The OHI score was five (see Figure 2).

## 4. Discussion

In this article, we describe multiple analytical strategies that were first developed for forensic purposes, on a set of three bone samples collected in 2011.

The selection of the bone element for molecular analysis is an important step, as DNA is not preserved equally in skeletal elements from different anatomical regions of the human body [21,22,23,24,25,26,27,28]. The bone element type that offers the highest DNA yield, especially in ancient skeletons, is the temporal bone, in particular the inner ear of the petrous bone [29,30,31,32,33,34]. However, the temporal bone is not always available for genetic testing because of historical interest or of practical or ethical reasons [35]. In such cases, other bones have to be selected for the genetic testing of aged skeletons; long bones (mainly femur) are preferred for analyses [4], while metacarpal and other short bone elements provide promising or even better results [36]. Therefore, several bones from the same skeleton should be collected whenever possible [4,21,22,23,24,25,26,27,28]. Regardless of the type of bone selected, the intrabone part plays an important role because of the high variability of DNA preservation observed not only between different bones, but also within an individual bone. A big difference between diaphysis and epiphysis of the long bones was observed [4,16,17,27,28] and, as shown in the vertebrae, different parts of the same bone yielded variable amounts of DNA, resulting in different STR typing success [37], which differs according to the ratio between the compact cortical bone tissue and the trabecular bone tissue located in the inner parts of the bones [61]. When exposed to harmful environmental conditions for long periods of time, compact cortical bone tissues and its DNA survive longer than trabecular bone tissues [4,16,17]. Thus, because the preservation of DNA in aged skeletons depends on many factors that are very complex, case-to-case strategies need to be implemented carefully. In addition, several DNA extraction protocols were developed in the last decade. Most of the protocols are based on mechanical pulverization of the bone sample, followed by Na_2_EDTA decalcification and lysis [47], while some authors suggest to omit the pulverization step, replacing it with a longer (up to five days) decalcification step for large fragments of bone [62]. In addition, protocols have been developed to perform decalcification and lysis in the same step [63,64]. Finally, even the organic DNA purification protocol with phenol/chloroform has been replaced by silica-based or magnetic bead-based procedures [4].

In the present research, we selected a single bone (patella) collected from the artificially mummified body of the Baron Pasquale Revoltella (1795–1869), as well as two femur segments that were hypothesized to belong to the Baron’s mother, Domenica Privato Revoltella (1775–1830). These bone elements, at the time of the exhumation (2011), were considered among the most reliable in terms of DNA recovery and quality for the genetic analyses available at that time.

As shown by visual and histological examinations, the preservation of the Baron’s bone was excellent, likely thanks to the mummification procedures coupled with favorable environmental conditions. Although no strict relationship exists between histological preservation and degradation level of the nucleic acids [31,58], high amounts of well-preserved DNA were yielded from the inner (trabecular) part of the patella, as shown by agarose gel electrophoresis and qPCR analysis. This sample, stored for seven months at −80 °C after the exhumation, confirmed the results of our preliminary Y-STR PCR-CE typing, and yielded a full autosomal STR PCR-CE profile as well. In addition, a full identity SNP profile was obtained using PCR-MPS. Finally, mtDNA was successfully typed both with PCR-CE and PCR-MPS technologies. Therefore, despite the skeletal remains being 142 years old, the DNA sample showed no degradation or inhibition issues; this provides further evidence that environmental conditions are the major factor in DNA preservation [1,5,8,22,31,35,65]. The only indication that the sample was ancient was the historical record. However, it is noteworthy that relevant levels of degradation occurred during the 10 years the sample was stored at −80 °C; this was indicated by both the decrement of the number of mtDNA molecules and the lack of the amplification of the long qPCR mitochondrial target. This result is in agreement with previous studies, which found that freezing did not eliminate DNA degradation issues [66].

In addition to the Baron’s remains, a metal box that likely contained the Baron’s mother’s remains was found in a niche next to the Baron’s grave. D.P.R. died in 1830 and, according to Baron’s will, her skeletal remains were transferred from the Municipal Cemetery of Santa Anna (Trieste) to the San Pasquale crypt in 1870. The samples from the trabecular residues of Baron’s mother’s femurs initially gave no results [50]. In contrast, the compact cortical bone tissue, we analyzed from the same two femurs ten years later, provided excellent and reliable genotyping data. In fact, samples RF2 and LF2 yielded a consensus profile for 10 out of 16 autosomal STR markers, and 80 out of 90 identity SNP markers, respectively. Finally, mtDNA control region analysis was successfully performed using two different technologies. These results highlight how crucial it is to sample the long bones correctly, because no genotyping data was obtained from the trabecular bone of the two femurs analyzed. Therefore, our results support previous data showing that the compact cortical bone has to be preferred in genetic studies [4,16,21,22,23,24,26,27,28]. However, we strongly recommend that more than one bone sample, including the temporal [29,30,31,32,33,34,35,36] and the metatarsals [36] bones, should be collected if available, both in forensic and archaeological casework. It is likely that even the magnetic bead-based protocol [47] we used for DNA purification in 2022 contributed to the successful outcome.

The availability of the genotypic data from the alleged mother–son pair prompted us to perform a kinship analysis. The analysis of the autosomal markers (STR and identity SNP) showed a cumulative LR of 9,129,302 (corresponding to a probability of maternity of 99.9999%). Further evidence for the maternal relationship was found through mtDNA control region sequencing, which highlighted the sharing of the same haplotype between the Baron and D.P.R. bone samples. There is no doubt that a mother–son pair was studied (and therefore that the skeletal remains found in the metal box near the Baron’s grave belonged to his mother).

Finally, the analysis of the haploid SNP markers allowed us to establish that the Baron belongs to the R1b (R1b-M343) haplogroup, which originated in South-East Asia and then spread to Eurasia and the Americas. Even the minimal haplotype, built by eight Y-STR markers, confirmed the Baron’s ancestry as Eurasian. Altogether, these genetic data are in agreement with the historical records of the archives of the Municipality of Venice, where the data of his paternal lineage were found up to the grand-parents (no data were found on his maternal lineage).

## 5. Conclusions

This casework represents a challenging trial to test forensic protocols on bone samples of historical interest. Essentially, optimization of the DNA extraction procedures, thanks to the implementation of the instruments in suitable facilities, allowed the genetic typing, even with the gold standard PCR-CE technology. It is also true, however, that PCR-MPS technology is an extraordinary tool when large sets of markers need to be analyzed simultaneously, such as for the identity SNP panel studied here. Nevertheless, both techniques require duplicate tests, as well as stringent precautions for preventing/identifying exogenous contaminations, which can potentially lead to misleading conclusions [3,5,8,25]. Thus, the results of this study support the finding that the methods commonly used in forensic genetics are also suitable for the analysis of historical remains. However, the current limit of this analysis is the high cost of the next generation sequencing technology.

## Figures and Tables

**Figure 1 genes-14-00851-f001:**
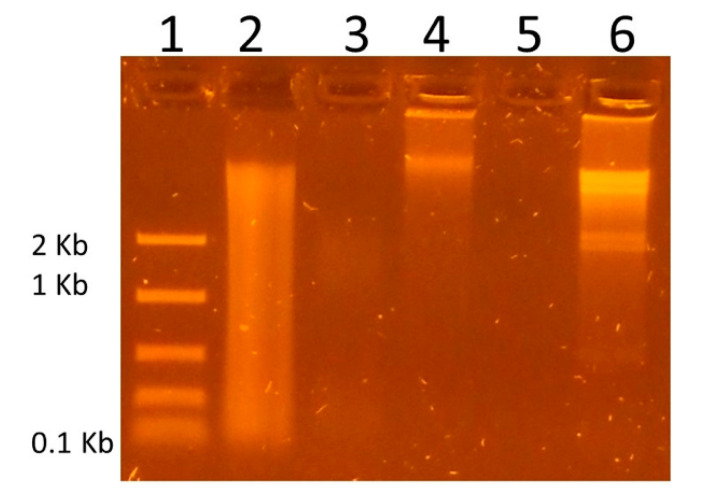
Agarose gel electrophoresis: 1 Easy Ladder; 2 sample P1; 3 external sample; 4 high molecular weight human DNA; 5 external sample; 6 lambda/Hind III molecular weight marker. Numbers on the left indicate reference Kilobases (Kb). Samples 3 and 5 are not included in this study.

**Figure 2 genes-14-00851-f002:**
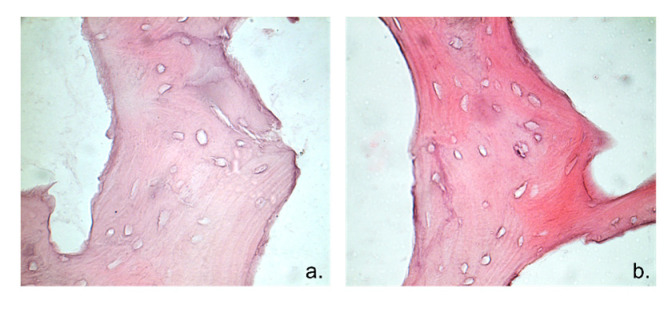
Histological images of two sections (**a**,**b**) of the patella bone. In both sections, details of bone trabeculae show well-preserved bone tissue with well-recognizable osteocyte lacunae and no signs of destructive foci (magnification 40×).

## Data Availability

Data is contained within the article or supplementary material. Other data presented in this study are available on request from the Corresponding Author.

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
