# Peer review of "The Baron Pasquale Revoltella’s Will in the Forensic Genetics Era"

_genes, 2023, doi:10.3390/genes14040851_

Round 1

Reviewer 1 Report

The revised manuscript concerns the application of a multidisciplinary approach to the analysis of the human remains of an Italian nobleman.  The article is interesting and presents an interesting case study in forensic archaeology. Such cases can be challenging and allow one to confront the advancement of modern techniques. The article is certainly worthy of publication after some issues have been ironed out. The introduction is too long, especially compared to the discussion: some parts could be removed from the beginning or added later. It is not clear to the reviewer what contribution the histological analysis of a portion of bone can make to the evaluation of the case. The authors refer to existing problems with DNA typing from bone and skeletal matrix. This aspect is fundamental in the study of bone remains and has important repercussions both forensically and archaeologically. Why wasn't a tooth, for example, or a rib used? There are some recent works that talk about how the yield of such matrices is optimal. I would invite the authors to review these aspects in order to implement the discussion. There are also papers that talk about bone extraction using LFD (large fragment demineralisation) from an Italian group, why was this not mentioned by the authors? Please include this aspect as well. The whole skeleton photo is not appropriate, photographs for forensic purposes must maintain standards of perspective, metric relief etc etc. I will be honoured to review the updated version of the manuscript.  

Author Response

See the attached file, please.

Reviewer 2 Report

The article is well written and addresses an interesting case of forensic archaeological analyses in a case of identification.

There are some inconsistencies that need to be reviewed.

-Why is the introduction so long? Some issues can be raised in the discussion part.

-The skeleton picture is not forensically appropriate.

- When dealing with skeletal remains, and this is the purpose also of the present special issue, genetic analyses play a pivotal role. Skeletal matrices can be very challenging and for that reason, recently, many different methodologies have been proposed. I’d like the authors to discuss the new chances in DNA extraction for example or the strategies to use in case of difficult samples or data interpretation.

Author Response

Please see the attachmet.

Round 2

Reviewer 1 Report

 I believe that the manuscript has been sufficiently improved to warrant publication in Genes.

Reviewer 2 Report

Authors provided all the requested changes.